# Demonstration of 144-Gbps Photonics-Assisted THz Wireless Transmission at 500 GHz Enabled by Joint DBN Equalizer

**DOI:** 10.3390/mi13101617

**Published:** 2022-09-27

**Authors:** Xiang Liu, Jiao Zhang, Shuang Gao, Weidong Tong, Yunwu Wang, Mingzheng Lei, Bingchang Hua, Yuancheng Cai, Yucong Zou, Min Zhu

**Affiliations:** 1National Mobile Communications Research Laboratory, Southeast University, Nanjing 210096, China; 2Purple Mountain Laboratories, Nanjing 211111, China

**Keywords:** Terahertz-band, multiple-input multiple-output, polarization multiplexing, seamless integration, Deep Belief Network

## Abstract

The THz wireless transmission system based on photonics has been a promising candidate for further 6G communication, which can provide hundreds of Gbps or even Tbps data capacity. In this paper, 144-Gbps dual polarization quadrature-phase-shift-keying (DP-QPSK) signal generation and transmission over a 20-km SSMF and 3-m wireless 2 × 2 multiple-input multiple-output (MIMO) link at 500 GHz have been demonstrated. To further compensate for the linear and nonlinear distortions during the fiber–wireless transmission, a novel joint Deep Belief Network (J-DBN) equalizer is proposed. Our proposed J-DBN-based schemes are mainly optimized based upon the constant modulus algorithm (CMA) and direct-detection least mean square (DD-LMS) equalization. The results indicate that the J-DBN equalizer has better bit error rate (BER) performance in receiver sensitivity. In addition, the computational complexity of the J-DBN-based equalizer can be approximately 46% lower than that of conventional equalizers with similar performance. To our knowledge, this is the first time that a novel joint DBN equalizer has been proposed based on classical algorithms. It is a promising scheme to meet the demands of future fiber–wireless integration communication for low power consumption, low cost, and high capacity.

## 1. Introduction

The explosive growth of wireless devices and streaming services, e.g., 3D video, cloud office, virtual reality (VR), internet of vehicles, and 6G services, has led to the demand for higher transmission capacity and more spectrum resources [1,2]. The Terahertz band (THz band), occupying the spectrum range of 0.3 THz to 10 THz, is a promising candidate for providing large capacity; it can provide a data capacity of hundreds of Gbps or even Tbps due to its substantial available bandwidth [3,4,5]. Moreover, the devices at THz band have a small and compact size, and can be monolithically integrated with other front-end circuits in portable terminals.

THz communication techniques can be divided into two main categories: pure electronic schemes and photonics-aided schemes. Photonics-aided THz-wave signals can be adopted in fiber–wireless communication systems, which overcomes the bandwidth limitation and electromagnetic interference resulting from electronic devices [6,7,8]. Moreover, the characteristics of photonic technology have effectively promoted the seamless integration of wireless and optical fiber networks. A photonics-assisted transmission system at the THz band integrates the large-capacity, long-distance advantages of fiber-optic transmission and a THz wireless transmission link, and therefore it has excellent potential for future 6G communication. In previous years, several seamless integration systems of THz wireless and optical fiber networks have been demonstrated, enabled by photonics [9,10,11,12,13,14,15]. However, the nonlinear impairments induced by photoelectric devices and wireless links will cause severe degradation and significantly restrict the transmission rate and distance. Therefore, advanced equalization algorithms are required to compensate for these nonlinear impairments.

To mitigate the distortions of nonlinearity, several nonlinear equalization methods have been widely researched, i.e., Volterra [16], Kernel [17], and Maximum Likelihood Sequence Estimator (MLSE) [18]. However, with the increase in transmission speed, conventional digital signal processing (DSP) algorithms cannot effectively equalize nonlinear noise such as relative intensity and modal partition noise in the practical system [19]. In addition, blind equalization, which performs adaptive channel equalization without the aid of a training sequence, has become a favorite of practitioners, such as the constant modulus algorithm (CMA) equalizer and direct-detection least mean square (DD-LMS) equalizer. As we know, CMA is a typical blind adaptive algorithm for adaptive blind equalizers in Quadrature Amplitude Modulation (QAM) format [20]. Nevertheless, CMA has a significant residual error after convergence, which is unsuitable for the nonlinear channel balance. Furthermore, more advanced artificial intelligence (AI) algorithms are considered as promising solutions for signal equalization in optical communication systems. Table 1 summarizes the typical photonics-aided wireless communication systems using AI equalization algorithms. Different paradigms based on artificial neural networks (CNN) [21,22] and deep neural networks (DNN) [23,24,25,26] have been proven to obtain better performance than conventional DSP algorithms, showing an excellent capability of mitigating nonlinear distortions. However, a CNN fails to fully account for the relevant characteristic information of the input sample, leading to performance degradation in the optical communication system [27]. Moreover, a CNN is unable to meet the demand for complex-valued equalization in wireless transmission systems [24]. Furthermore, a DNN requires many training sequences in order to achieve the optimum result, reducing the effective channel capacity. Due to the ability to learn the relevant information and make the nonlinear decision in feature space, the Deep Belief Network (DBN) is used for nonlinear equalization, which is more helpful in handling the receiver sensitivity issue in the THz-band wireless communication system [28].

In this study, we propose a novel joint DBN (J-DBN) equalizer for the combination of two loss functions based on blind CMA and DD-LMS equalizers. The J-DBN nonlinear equalizer not only retains an excellent capability of mitigating nonlinear impairment but also reduces the computational complexity. By using our proposed J-DNN equalizer, we experimentally off-line demonstrated a 144 Gbit/s dual polarization QPSK (DP-QPSK) signal generation over a 20-km standard single-mode fiber (SSMF) and 3-m wireless 2 × 2 multiple-input multiple-output (MIMO) link at 500 GHz with BER below 3.8 × 10^−3^. The results show that a J-DBN equalizer can significantly improve the receiver sensitivity performance. In addition, the computational complexity of the J-DBN equalizer can be approximately 46% lower than that of traditional equalizers with similar BER performance.

## 2. Operation Principle for J-DBN-Based Equalization

### 2.1. Traditional DBN Method

The DBN can be approximated as a stack of restricted Boltzmann machines (RBMs) [28], as shown in Figure 1a. An RBM is a generative stochastic network containing a visible layer $v={\left\{v_{i}\right\}}_{i=1}^{N}$ and a hidden layer $h={\left\{h_{j}\right\}}_{j=1}^{M}$ with the parameters θ={W,d,c}. The energy function and the likelihood function of the RBM can be stated as
(1)$$E(v,h)=-\sum_{i,j}v_{i}W_{ij}h_{j}-\sum_{i}d_{i}v_{i}-\sum_{j}c_{j}h_{j},$$
(2)$$P\left(v,h\right)=\frac{1}{Z}\exp(-E(v,h)),$$
where $v_{i}\in \left\{0,1\right\}$, $h_{j}\in \left\{0,1\right\}$, $W=\left(W_{ij}\right)\in R^{N\times M}$ are the weights connecting the visible layer and hidden layer, *d* and *c* are the bias terms of the visible and hidden layers, and *Z* represents the partition function. Moreover, the probabilities P(v|h) and P(h|v) can be calculated by
(3)$$P(v_{i}=1|h)=\sigma \left(\sum_{j}W_{ij}h_{j}+d_{i}\right),$$
(4)$$P(h_{j}=1|v)=\sigma \left(\sum_{i}W_{ij}v_{i}+c_{j}\right),$$
where σ(⋅) is the sigmoid function defined as $\sigma (x)=1/(1+e^{-x})$.

The DBN with λ hidden layers contains $W_{1},W_{2},\ldots ,W_{\lambda }$ connection weight matrices and λ+1 biases $d_{0},d_{1},\ldots ,d_{\lambda }$, where $d_{0}$ is the bias of the visible layer v. Therefore, the output probability can be calculated by the hidden vector: $\Phi =\sigma \left(W_{\lambda }h_{\lambda -1}+d_{\lambda }\right).$

Figure 1b,c represent the *j*-th neuron of the feedforward and reconstruction architecture. The RBM utilizes the stochastic approximation method to update the parameters θ={W,d,c} by maximizing the likelihood $P_{\theta }\left(v\right)$.

### 2.2. J-DBN Nonlinear Equalizers

There are two main NN-based equalization schemes, namely the blind NN equalization and adaptive NN equalization algorithms. To avoid the residual error for equalizing nonlinear channels, the CMA equalizer can be combined with the NN algorithm. Meanwhile, DD-LMS combined with the NN equalization algorithm can further accelerate the convergence speed and ensure the optimum nonlinear decision. Our proposed J-DBN method, including an adaptive DBN-1-based equalizer and a blind DBN-2-based equalizer, is mainly based upon CMA and DD-LMS blind equalizers, as shown in Figure 2a.

Figure 2b shows the detailed architecture of our proposed J-DBN equalizer, including two sequential DBNs. Each link between one visible layer and multiple hidden layers in J-DBN is associated with the weight value $w_{ij}^{k}$, where *k* denotes the *k*-th hidden layer, and *i* and *j* represent the *i*-th node in the visible layer and the *j*-th node in the current hidden layer, respectively. The output of nonlinear neurons is summed as $h_{j}^{k}=\sigma \left(\sum_{i=1}^{S}w_{ij}^{k}x_{i}\right)$, where *S* is defined as the length of input samples, and σ(⋅) denotes the nonlinear active function between multiple hidden layers. The selection of a matching activation function is an important part of DBN construction.

During the feedforward training process, the output of the *j*-th neuron in the other (*k +* 1)-th visible layer can be calculated as $v_{j}^{k+1}=\sigma \left(\sum_{i=1}^{N}w_{ij}^{k+1}v_{j}^{k}\right)$, where $v_{j}^{k}$ is the output of the *j*-th neuron cell in the *k*th hidden layer, and the total number of cells in this layer is *N*. When *k* is increased as *λ*, we define $h_{j}^{L+1}$ as the final output signal calculated from the output layer. The DBN-1 and DBN-2 equalizers update the weights according to their respective cost functions, guaranteeing the optimal solution of the whole equalization process.

### 2.3. Cost Function of DBN-1 and DBN-2 Equalizers

#### 2.3.1. Cost Function of the Adaptive DBN-1-Based Equalizer

As we know, CMA is a typical blind equalizer for OOK or QPSK modulation format with a single reference radius. To reduce the large residual error after convergence, CMA blind equalization can be combined with a DBN-1 equalizer. The corresponding filter tap weight $h_{nn}$ and the cost function of the DBN-1 equalizer based on the CMA algorithm are defined as
(5)$$h_{nn}\to h_{nn}+\mu e_{n\_CMA}x\left(n\right)$$
(6)$$J_{n\_CMA}=\frac{1}{2}\sum_{n=1}^{Q}{\left(R_{1}-\left|O\left(n\right)\right|\right)}^{2}$$
where *μ* is defined as the CMA convergence parameter, *R*_1_ is the constant module of signals, O(n) is the output of the DBN-1 network, and the input sequence *x*(*n*) in the optimization network is the test signal with a size of *Q.*

The adaptive error function can be defined as $e_{n\_CMA}=R_{1}-\left|O(n)\right|$. In practical terms, we can find the optimum *μ* and tap number to obtain better channel equalization. Moreover, the weight value can be further optimized with the aid of adaptive equalization and the BP algorithm, which can be given as
(7)$$w_{ij}^{k+}=w_{ij}^{k}-\Delta w_{ij}^{k}=w_{ij}^{k}-\eta \frac{\partial J_{n\_CMA}}{\partial w_{ij}^{k}}$$

#### 2.3.2. Cost Function of Blind DBN-2-Based Equalizer

Unlike forwarding propagation steps, the weight values and model hyper-parameters are updated based on the backpropagation (BP) algorithm. According to the minimum mean square error (MMSE) algorithm, the cost function of the DBN-2 blind equalizer based on the DD-LMS algorithm can be defined as
(8)$$J_{n\_ddlms}=\frac{1}{2}\sum_{n=1}^{S}{\left(T_{n}-O_{2}\left(n\right)\right)}^{2}$$

Compared the obtained value $O_{2}(n)$ with the corresponding expected value $T_{n}$, the error $e_{n}=T_{n}-O_{2}(n)$ is sent to the network with a reverse training algorithm. Thus, the connected weight vector $w_{ij}^{k+}$ can be iteratively updated until the desired epoch or error value is reached, which can be represented as
(9)$$w_{ij}^{k+}=w_{ij}^{k}-\Delta w_{ij}^{k}=w_{ij}^{k}-\eta \frac{\partial J_{n\_ddlms}}{\partial w_{ij}^{k}}$$
where *η* is the learning rate and Δ represents the gradient operation. In order to accurately update the weight value of every nonlinear node, we must calculate the gradient of the whole training sequence. Owing to the introduction of the blind error function, we will decrease the training size and improve the computation speed effectively.

Thus, our proposed J-DBN equalizer comprises two parts, including the DBN-1 adaptive equalization and the DBN-2 blind equalization. Note that the former updating of the weight value is based on the traditional BP algorithm, while the latter deploys the blind equalization algorithm to optimize the network further.

## 3. Experimental Setup

In our previous work, we have established some real-time photonics-aided THz seamless integration transmission systems [12,13,14,15]. In order to verify the effectiveness of our proposed algorithms in this paper, we further perform an experimental demonstration for 144−Gbps photonics-assisted THz wireless transmission at 500 GHz enabled by J-DBN equalization. A detailed description of the experimental setup is shown in Figure 3, including the optical and THz transmitter modules, THz 2 × 2 MIMO wireless link, THz receiver module, and the off-line DSP blocks. For a fair comparison, there were four alternative algorithms included in the experiment at the off-line DSP blocks.

### 3.1. Optical and THz Transmitter Modules

Figure 3 provides the experimental setup of the optical and THz transmitter modules. An arbitrary waveform generator (AWG) of 92-Gsa/s sampling rate is used to generate the I and Q components of the baseband electrical signals. A parallel electrical amplifier (EAs) is used to amplify the I/Q electrical signals. Then, a 193.5-THz linewidth external cavity laser-1 (ECL-1) is used to produce the optical carrier, which is modulated via an I/Q modulator with 30-GHz bandwidth and approximately 7-dB insertion loss. The modulated optical signal is divided to the polarization multiplexing channels by an optical coupler (OC) to simulate signal delay and attenuation. One is transmitted through a 1-m fiber direct link (DL) and the other passes through a variable optical attenuator (VOA). Then, after a polarization beam coupler (PBC), the 193.5-THz optical baseband signal is sent to SSMF.

Erbium-doped fiber amplifiers (EDFA) are used to compensate for the optical-fiber transmission loss after a 20-km SSMF transmission link. In order to suppress the out-of-band amplified spontaneous emission (ASE) noise, a passband tunable optical filter (TOF) is used. A free-running tunable external cavity laser (ECL-2) is operated as an optical local oscillator (LO), which has a linewidth of less than 100 kHz. An optical baseband signal with 10.5 dBm optical power and an optical LO with 13.5 dBm optical power are coupled by an OC. The optical spectra of the optical signal with the tunable optical LO after OC (0.03 nm resolution) are shown in Figure 3b. Note that the optical power of the X- and Y-polarization components after PBS should be as equal as possible. The AIPMs used in our setup are polarization-sensitive, with a maximum of 4.5 dB polarization-dependent responsivity (PDR). The X- and Y-polarization imbalance will result in 2 × 2 MIMO THz-wave imbalance and deteriorate the system performance. Therefore, two polarization controllers (PCs) are required before the OC. In the test, the optical signal and optical LO separately adjust the incident X- and Y-polarization direction to maximize the optical power in the antenna-integrated photomixer module (AIPM, NTT Electronics Corp. IOD-PMAN-13001).

The side-mode suppression ratio (SMSR) of the optical signal and optical LO is >50 dB. In our proposed system, THz-wave wireless signals with a tunable carrier frequency range from 340 GHz to 530 GHz are generated by photonic heterodyning using AIPMs. The AIPM consists of an ultra-fast uni-traveling-carrier photodiode (UTC-PD) and a bow-tie or log-periodic antenna. Two parallel AIPMs are used, each with a typical −28 dBm output power and operating wavelength range from 1540 nm to 1560 nm. The typical photodiode responsivity is 0.15 A/W, and the maximum optical input power is 15 dBm. PBS is used to separate the X- and Y-polarization components by APIMs to generate two parallel THz-wave wireless signals, respectively. To drive the AIPMs, another EDFA is used to boost the optical power of combined lightwaves before PBS.

### 3.2. THz 2 × 2 MIMO Wireless Link and THz Receiver Modules

Figure 3 shows the THz 2 × 2 MIMO wireless transmission link and lens position. The two parallel THz-wave signals from the AIPMs are transmitted over a 3-m 2 × 2 MIMO wireless transmission link. In order to focus the wireless THz-wave, three pairs of lenses are deployed to maximize the received THz-wave signal power and are manually aligned. Lenses 1, 2, 5 and lenses 3, 4, 6 are aligned with the X-polarization and Y-polarization wireless link, respectively. Lenses 1–4 are identical, each having a 20-cm focal length and 10-cm diameter. The smaller lenses 5 and 6 have a 10-cm focal length and 5-cm diameter, which are used for THz-waves’ high-accuracy alignment to horn antennas (HA). For the X-polarization (Y-polarization) THz wireless link, the longitudinal separation distance between the AIPM and lens 1 (lens 3), lens 2 (lens 4), and lens 5 (lens 6) and the receiver HA are 0.2 m, 3 m, and 5 cm, respectively. The lateral separation between the two AIPMs and two HA pairs is 25 cm. In order to avoid multi-path fading from reflections on the optical table, the conversion modules are placed at the height of 20 cm. Photos of the 3-m wireless transmission link and lens position are shown in Insets Figure 3c and Figure 3d, respectively.

At the THz-wave receiver end, THz-wave wireless signals are received with two parallel THz-band HAs with 26-dBi gain. Electronic LO sources drive two identical THz receivers to implement analog down-conversion for X- and Y-polarization THz-wave wireless signals. Each consists of a mixer, a ×12 frequency multiplier chain, and an amplifier, and operates at 500 GHz. Then, the down-converted X- and Y-polarization intermediate-frequency (IF) signal at 20 GHz is boosted by two cascaded electrical low-noise amplifiers (LNAs) with a 3-dB bandwidth of 47 GHz and captured by a digital oscilloscope with a 128-Gsa/s sampling rate.

### 3.3. Off-Line DSP Blocks

The block diagram of the off-line DSP is illustrated in Figure 3a,e. In the Tx-side DSP, a QPSK symbol mapping and a raised-cosine (RC) filter with a roll-off factor of 0.01 are deployed, as shown in Figure 3a. Figure 3e depicts the four DSP options at the Rx side, verifying the validity of the J-DBN-based equalization schemes. Conventional, Opt. 1: the captured signal is offline, proposed by typical DSP steps including down-conversion into baseband, resampling, Gram–Schmidt orthogonalization process (GSOP), followed by 53-tap CMA equalization. Then, the frequency offset noise can be mitigated via frequency offset estimation (FOE), and the phase offset problem can be solved after carrier phase recovery (CPR). Finally, a 37-tap DD-LMS equalizer is added to compensate for the remaining linear damage and I/Q imbalance before BER calculation.

We also compare the BER performance between the CMA equalizer, DBN-based equalizer, and J-DBN equalizer within these DSP steps. In Opt. 2, the adaptive DBN-1 equalizer adapts itself to compensate for the nonlinear distortion. It extracts the signal sequences’ characteristics based on the BP algorithm and blind CMA algorithm. The scheme can reduce the sizeable residual error after convergence, which is suitable for the nonlinear channel balance. In Opt. 3, the blind DBN-2 equalizer optimizes the weight value and tap number based on the BP algorithm and DD-LMS equalizer. Moreover, the DBN-2 equalizer can utilize the weight value updated by the DBN-1 as the initial value. The scheme can further compensate for the remaining linear damage and nonlinear decision ability. In Opt. 4, the J-DBN equalizer combines two error cost functions, which is more helpful in handling the receiver sensitivity issue in THz-band wireless links and achieving a more accurate BER decision. Moreover, the J-DBN equalizer can be established via two steps. Firstly, it can be initialized with the aid of the training sequence, and then the weight value can be further optimized by employing the error function of CMA. Adopting our proposed J-DBN equalizer, both the lengths of the training data and the training time can be effectively reduced.

## 4. Experimental Results and Discussion

As we know, a well-designed equalizer is useful for resolving nonlinear issues and has been successfully applied in wireless communications. However, the selection of an error function is an important factor that affects the residual error of the blind equalizer; thus, the performance of the equalizer is different. In our experiment, we introduce some equalizers in our proposed schemes and compare their performance, such as the typical CMA equalizer with taps and the DD-LMS equalizer.

### 4.1. Transmission Results

We first measure the performance of our transparent fiber-optical and THz wireless 2 × 2 MIMO transmission system for the back-to-back (BtB) case, i.e., without fiber and wireless distance transmission. Figure 4 gives the measured BER of X- and Y-polarization versus different input power into each AIPM. At the 3.8 × 10^−3^ HD-FEC limit, the THz-wave carrier frequency at 500 GHz for the BtB case has a successfully transmitted range from 28 GBaud to 36 GBaud. The insets show that the QPSK constellation points can be demodulated well at 28 GBaud. However, the BER performance degrades at high transmission rates due to the limitations of the receiver LNA bandwidth.

Then, we measure the BER versus the input power into each AIPM over one span of 20-km SSMF and 3-m wireless distance, as shown in Figure 5. At the 3.8 × 10^−3^ HD-FEC limit, the THz-wave carrier frequency at 500 GHz for the fiber and wireless transmission can also be successfully transmitted. However, the transmission performance is not ideal at 36 GBaud. Figure 6 gives the electrical spectrum of the 24/28/32/36 GBaud QPSK IF signal with the corresponding bandwidth (BW). The signals are more damaged when the transmission speed increases. Moreover, we can see that the required bandwidth becomes limited as the transmission rate increases, especially at the 36 GBaud rate with 36.36 GHz bandwidth. To protect the AIPMs, the maximum input optical power is set at 13.5 dBm. A THz-wave carrier frequency at 500 GHz can be successfully transmitted at the 3.8 × 10^−3^ HD-FEC threshold. The best BER performance occurs for the lower transmission rates. The insets also show that the constellation points can be demodulated well. In order to further improve the performance for higher transmission rates, the J-DBN equalizer based on the conventional DSP algorithm is introduced.

Here, the training data are only used for the DBN-1 adaptive equalizer in our proposed J-DBN equalizer since the DBN-2 blind equalizer and optimization is a self-recovering equalization method without the aid of a training sequence. It indicates that the J-DBN equalizer has good training accuracy and satisfactory tracking speed. Figure 7 illustrates the BER of 36 GBaud QPSK signals versus the input optical power into each AIPM and SNR over one span of 20-km SSMF and 3-m wireless distance; it can be found that increasing the optical power can help to improve the BER performance due to the larger SNR. Next, we compare four equalization schemes (including *Opt*. 1–*Opt*. 4) with a 53-tap CMA equalizer, 37-tap DD-LMS equalizer, DBN-1 equalizer with 25 cells, and DBN-2 equalizer with 50 cells. From the comparison between *Opt*. 1 and *Opt*. 4, it can be concluded that the required power under HD-FEC (3.8 × 10^−3^) utilizing the J-DBN scheme is close to 12.6 dBm, which increases by almost up to 0.2 dB in receiver sensitivity and 0.8 dB in SNR gain compared with the conventional scheme. Moreover, *Opt*. 2 and *Opt*. 3 also obtain a slight performance improvement after the DBN-1 equalizer or DBN-2 equalizer. When the input power into the AIPM is 12.8 dBm, the illustrations depict the constellation diagrams of QPSK symbols after recovery. The received QPSK symbol constellation before the receiver DSP chain is also given. Furthermore, we compared the constellation diagrams of the *Opt*. 1 and *Opt*. 4 schemes. The illustrations (i) and (ii) also show that the J-DBN equalizer can reduce the residual error after convergence and visually improve the nonlinear decision capacity, which is suitable for the nonlinear channel balance.

### 4.2. Complexity Analysis

We further analyze the complexity of the proposed J-DBN nonlinear equalizer and make a comparison with blind CMA and DD-LMS equalizers. We consider the complexity in two aspects, convergence steps and computation time, as is shown in Figure 8. Figure 8a shows that the convergence speeds of the DBN-1 and DBN-2 equalizers are faster than the conventional CMA and DD-LMS equalizers, which verifies the well-trained neural networks. Moreover, the computation time of DBN-based equalizers can be approximately 36.3% and 46% lower than that of CMA and DD-LMS-based methods with similar BER performance, as is shown in Figure 8b. Therefore, the conclusion can be reached that the J-DBN equalizer has a significant advantage in dealing with nonlinear distortion, making it quite suitable in high-speed THz-band wireless communication systems.

## 5. Conclusions

In this paper, a novel J-DBN equalizer for the 144-Gbps QPSK signal transmission system over a 20-km SSMF and 3-m THz-band wireless link at 500-GHz is experimentally demonstrated, which consists of two steps including a DBN-1 adaptive equalizer based on the CMA algorithm and a DBN-2 blind equalizer based on the DD-LMS algorithm. We compare the J-DBN equalizer with the classical equalizer in terms of the BER performance. Meanwhile, our proposed J-DBN equalizer in the adaptive equalization step can reduce the computational complexity and obtain better training accuracy during the self-recovering blind equalization process. The experimental results show that the J-DBN method has good training accuracy, a smaller requirement for training sequences, and satisfactory computational complexity. Thanks to our proposed J-DBN scheme, an improvement of 0.2 dB and 0.8 dB over the J-DBN equalizer in receiver sensitivity and SNR gains at a BER of 3.8 × 10^−3^ is achieved compared with conventional equalizers. Moreover, the computational time is effectively improved, being up to 46% lower than the conventional equalizer. To our knowledge, this is the first time that joint DBN equalizers have been deployed in the THz-band wireless transmission link. Our proposed J-DBN equalization scheme is promising for future 6G fiber-optical and THz-wireless seamless integration communication systems.

## Figures and Tables

**Figure 1 micromachines-13-01617-f001:**
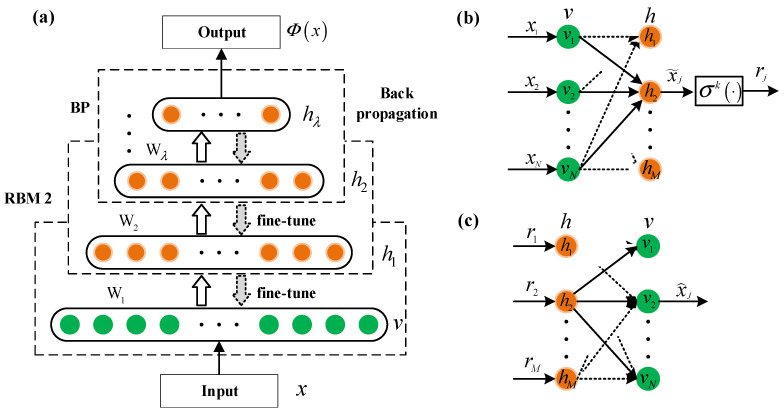
(**a**) The structure of DBN, which can be approximated as a stack of RBMs. (**b**) The feedforward architecture of the *j*-th hidden neuron. (**c**) The reconstruction architecture of the *j*-th hidden neuron.

**Figure 2 micromachines-13-01617-f002:**
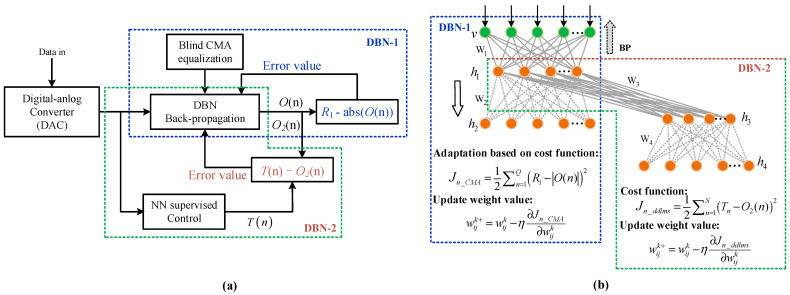
The proposed J-DBN equalizers: (**a**) schematic diagram; (**b**) architecture.

**Figure 3 micromachines-13-01617-f003:**
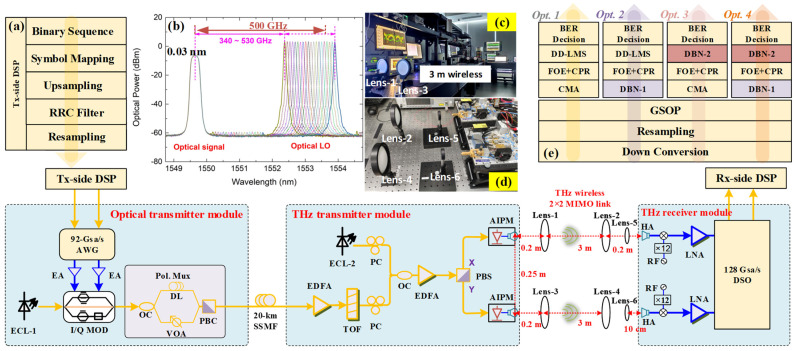
Experimental setup of photonics−aided THz wireless transmission system over 20-km SSMF and 3-m wireless distance with detailed DSP blocks at Tx- and Rx-side, including (**a**) Tx-side DSP block, (**b**) the optical spectra of the optical signal with tunable optical LO after optical coupler (0.03 nm resolution). Insets: (**c**) 3-m 2 × 2 MIMO wireless transmission link. (**d**) Lens position at THz receiver side. (**e**) Rx-side DSP block with three proposed J−DBN optional schemes.

**Figure 4 micromachines-13-01617-f004:**
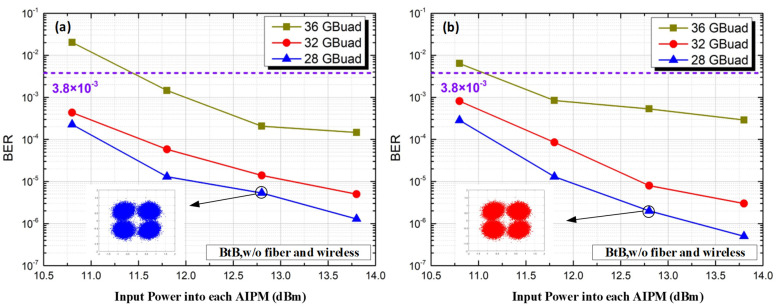
BER versus input power into each AIPM for the BtB case without fiber and wireless transmission. (**a**) X-polarization; (**b**) Y-polarization.

**Figure 5 micromachines-13-01617-f005:**
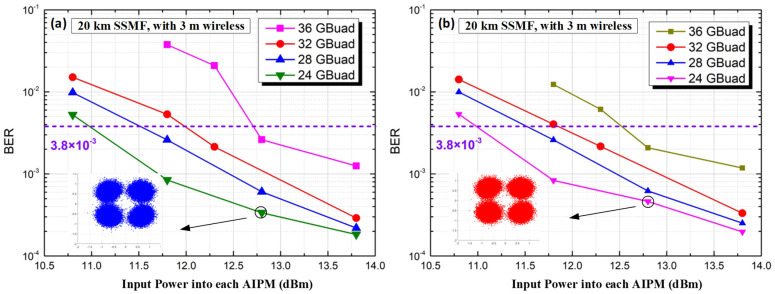
BER versus input power into each AIPM over one span of 20-km SSMF and 3-m wireless transmission. (**a**) X-polarization; (**b**) Y-polarization.

**Figure 6 micromachines-13-01617-f006:**
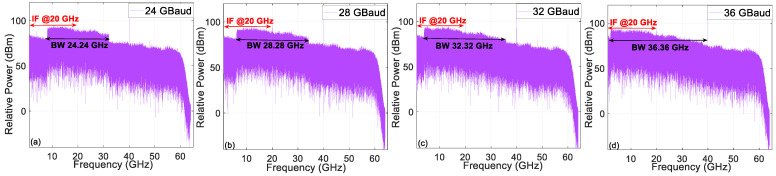
Electrical spectrum of the received QPSK IF signal: (**a**) 24 GBaud signal with 24.24 GHz BW; (**b**) 28 GBaud signal with 28.28 GHz BW; (**c**) 32 GBaud signal with 32.32 GHz BW; (**d**) 36 GBaud signal with 36.36 GHz BW.

**Figure 7 micromachines-13-01617-f007:**
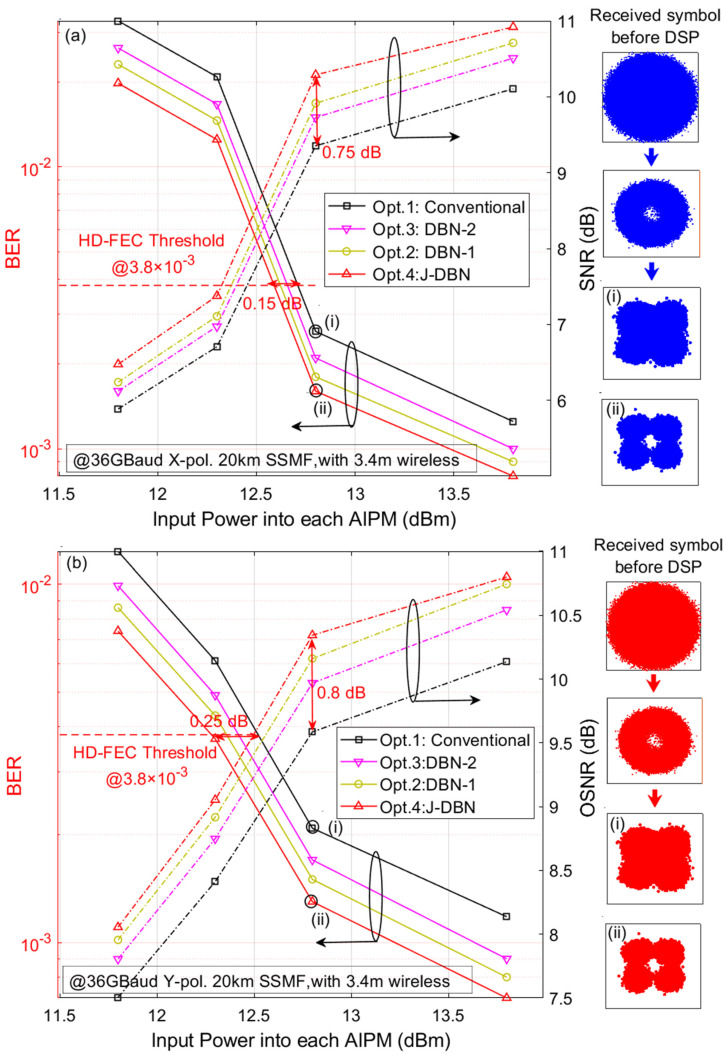
BER versus input power into each AIPM over one span of 20-km SSMF and 3-m wireless transmission for different proposed equalizers. (**a**) X-polarization. (**b**) Y-polarization.

**Figure 8 micromachines-13-01617-f008:**
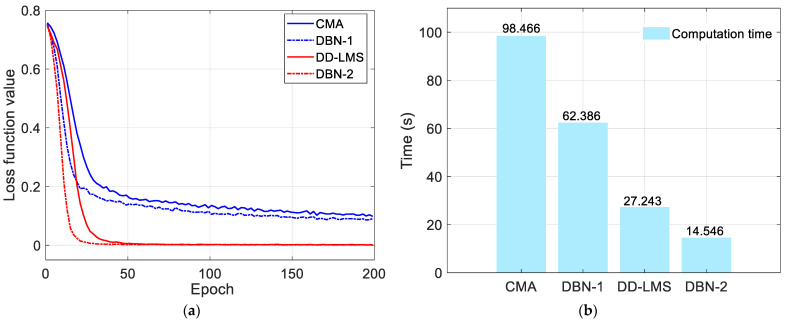
Complexity analysis of the 36 Gbaud QPSK signals with different equalization options when the input power into AIPM is 12.8 dBm. (**a**) Loss function versus iteration numbers. (**b**) Comparison of the computation time.

**Table 1 micromachines-13-01617-t001:** Summary of photonics-aided wireless transmission systems based on AI algorithms.

Frequency (GHz)	Line Rate (Gb/s)	Format	Distance (m)	Pol. Mux	AI Algorithm	Year	Ref.
340	53.5	16QAM	54.6	Single	CNN	2022	[21]
120	40/55	16/64QAM	200	Single	2D-CNN	2022	[22]
135	60	PAM-8	3	Single	DNN	2020	[23]
140	90	PAM-4	3	Single	DNN/LSTM	2021	[24]
80	60	64QAM	1.2	Single	Dual-GRU	2022	[25]
500	144	QPSK	3	Dual	DBN	2022	This work

## Data Availability

Not applicable.

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
