# Peer review of "Demonstration of 144-Gbps Photonics-Assisted THz Wireless Transmission at 500 GHz Enabled by Joint DBN Equalizer"

_micromachines, 2022, doi:10.3390/mi13101617_

Round 1
Reviewer 1 Report
Authors propose a novel joint Deep Belief Network (J-DBN) equalizer, in order to compensate for the linear and nonlinear distortions during the fiber-wireless transmission. Their proposed J-DBN-based schemes are optimized towards constant modulus algorithm (CMA) and direct-detection least mean square (DD-LMS) equalization. The results obtained indicate a better bit error rate (BER) performance in receiver sensitivity, and the computational complexity is about 46% lower than that of conventional equalizers with a similar performance. The work represents a promising scheme to meet the demands of future fiber-wireless-integrations communication for low power consumption, low cost, and high capacity. I therefore strongly recommend the proposed paper for publication in MDPI Micromachines.
Reviewer 2 Report
The paper presents 144 Gbps bitrate over fiber using a subcarrier at 500 GHz. The experimental work is defnitely commendable. Dual polarization is used and a wireless link is also demostrated.
The first thing that comes to mind is that how DSP can run at this speed. Then authors explain it is OFF-LINE DSP. They should mention this in the itroduction section. Off-line DSP cannot be used in real time communication and this is a major issue. Other than these issues the work is excellent.
